# Learning to Propagate Labels: Transductive Propagation Network for Few-shot Learning

**Yanbin Liu**[1,*] **Juho Lee**[2,3]**, Minseop Park**[3]**, Saehoon Kim**[3]**, Eunho Yang**[3,4]**,**
**Sung Ju Hwang**[3,4] **& Yi Yang**[1,5,†]
[1]CAI, University of Technology Sydney, [2]University of Oxford
[3]AITRICS, [4]KAIST, [5]Baidu Research
`csyanbin@gmail.com, juho.lee@stats.ox.ac.uk,`
`{mike_seop, shkim}@aitrics.com, {eunhoy, sjhwang82}@kaist.ac.kr,`
`Yi.Yang@uts.edu.au`

## Abstract

The goal of few-shot learning is to learn a classifier that generalizes well even when trained with a limited number of training instances per class. The recently introduced meta-learning approaches tackle this problem by learning a generic classifier across a large number of multiclass classification tasks and generalizing the model to a new task. Yet, even with such meta-learning, the low-data problem in the novel classification task still remains. In this paper, we propose *Transductive Propagation Network* (TPN), a novel meta-learning framework for transductive inference that classifies the entire test set at once to alleviate the low-data problem. Specifically, we propose to *learn to propagate labels* from labeled instances to unlabeled test instances, by learning a graph construction module that exploits the manifold structure in the data. TPN jointly learns both the parameters of feature embedding and the graph construction in an end-to-end manner. We validate TPN on multiple benchmark datasets, on which it largely outperforms existing few-shot learning approaches and achieves the state-of-the-art results.

## 1 Introduction

Recent breakthroughs in deep learning (Krizhevsky *et al.*, 2012; Simonyan and Zisserman, 2015; He *et al.*, 2016) highly rely on the availability of large amounts of labeled data. However, this reliance on large data increases the burden of data collection, which hinders its potential applications to the low-data regime where the labeled data is rare and difficult to gather. On the contrary, humans have the ability to recognize new objects after observing only one or few instances (Lake *et al.*, 2011). For example, children can generalize the concept of "apple" after given a single instance of it. This significant gap between human and deep learning has reawakened the research interest on few-shot learning (Vinyals *et al.*, 2016; Snell *et al.*, 2017; Finn *et al.*, 2017; Ravi and Larochelle, 2017; Lee and Choi, 2018; Xu *et al.*, 2017; Wang *et al.*, 2018).

Few-shot learning aims to learn a classifier that generalizes well with a few examples of each of these classes. Traditional techniques such as fine-tuning (Jia *et al.*, 2014) that work well with deep learning models would severely overfit on this task (Vinyals *et al.*, 2016; Finn *et al.*, 2017), since a single or only a few labeled instances would not accurately represent the true data distribution and will result in learning classifiers with high variance, which will not generalize well to new data.

In order to solve this overfitting problem, Vinyals *et al.* (2016) proposed a meta-learning strategy which learns over diverse classification tasks over large number of episodes rather than only on the target classification task. In each episode, the algorithm learns the embedding of the few labeled examples (the *support set*), which can be used to predict classes for the unlabeled points (the *query set*) by distance in the embedding space. The purpose of episodic training is to mimic

---

[*]This work was done when Yanbin Liu was an intern at AITRICS.

[†]Part of this work was done when Yi Yang was visiting Baidu Research during his Professional Experience Program.

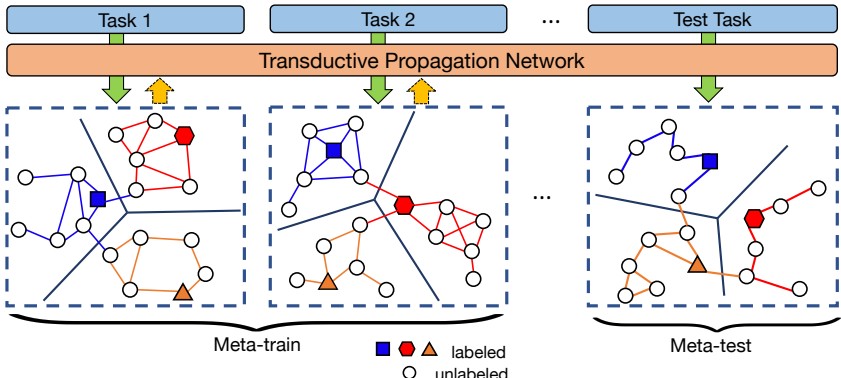

Figure 1: A conceptual illustration of our transductive meta-learning framework, where lines between nodes represent graph connections and their colors represent the potential direction of label propagation. The neighborhood graph is episodic-wisely trained for transductive inference.

the real test environment containing few-shot support set and unlabeled query set. The consistency between training and test environment alleviates the distribution gap and improves generalization. This episodic meta-learning strategy, due to its generalization performance, has been adapted by many follow-up work on few-shot learning. Finn *et al.* (2017) learned a good initialization that can adapt quickly to the target tasks. Snell *et al.* (2017) used episodes to train a good representation and predict classes by computing Euclidean distance with respect to class prototypes.

Although episodic strategy is an effective approach for few-shot learning as it aims at generalizing to unseen classification tasks, the fundamental difficulty with learning with scarce data remains for a novel classification task. One way to achieve larger improvements with limited amount of training data is to consider relationships between instances in the test set and thus predicting them as a whole, which is referred to as transduction, or transductive inference. In previous work (Joachims, 1999; Zhou *et al.*, 2004; Vapnik, 1999), transductive inference has shown to outperform inductive methods which predict test examples one by one, especially in small training sets. One popular approach for transduction is to construct a network on both the labeled and unlabeled data, and propagate labels between them for joint prediction. However, the main challenge with such label propagation (and transduction) is that the label propagation network is often obtained without consideration of the main task, since it is not possible to learn them at the test time.

Yet, with the meta-learning by episodic training, we can learn the label propagation network as the query examples sampled from the training set can be used to simulate the real test set for transductive inference. Motivated by this finding, we propose *Transductive Propagation Network* (TPN) to deal with the low-data problem. Instead of applying the inductive inference, we utilize the entire query set for transductive inference (see Figure 1). Specifically, we first map the input to an embedding space using a deep neural network. Then a graph construction module is proposed to exploit the manifold structure of the novel class space using the union of support set and query set. According to the graph structure, iterative label propagation is applied to propagate labels from the support set to the query set and finally leads to a closed-form solution. With the propagated scores and ground truth labels of the query set, we compute the cross-entropy loss with respect to the feature embedding and graph construction parameters. Finally, all parameters can be updated end-to-end using backpropagation.

The main contribution of this work is threefold.

- To the best of our knowledge, we are the first to model transductive inference explicitly in few-shot learning. Although Nichol *et al.* (2018) experimented with a transductive setting, they only share information between test examples by batch normalization rather than directly proposing a transductive model.

- In transductive inference, we propose to *learn to propagate labels* between data instances for unseen classes via episodic meta-learning. This learned label propagation graph is

shown to significantly outperform naive heuristic-based label propagation methods (Zhou *et al.*, 2004).

- We evaluate our approach on two benchmark datasets for few-shot learning, namely *mini*ImageNet and *tiered*ImageNet. The experimental results show that our *Transductive Propagation Network* outperforms the state-of-the-art methods on both datasets. Also, with semi-supervised learning, our algorithm achieves even higher performance, outperforming all semi-supervised few-shot learning baselines.

## 2 RELATED WORK

**Meta-learning** In recent works, few-shot learning often follows the idea of meta-learning (Schmidhuber, 1987; Thrun and Pratt, 2012). Meta-learning tries to optimize over batches of tasks rather than batches of data points. Each task corresponds to a learning problem, obtaining good performance on these tasks helps to learn quickly and generalize well to the target few-shot problem without suffering from overfitting. The well-known MAML approach (Finn *et al.*, 2017) aims to find more transferable representations with sensitive parameters. A first-order meta-learning approach named Reptile is proposed by Nichol *et al.* (2018). It is closely related to first-order MAML but does not need a training-test split for each task. Compared with the above methods, our algorithm has a closed-form solution for label propagation on the query points, thus avoiding gradient computation in the inner updateand usually performs more efficiently.

**Embedding and metric learning approaches** Another category of few-shot learning approach aims to optimize the transferable embedding using metric learning approaches. Matching networks (Vinyals *et al.*, 2016) produce a weighted nearest neighbor classifier given the support set and adjust feature embedding according to the performance on the query set. Prototypical networks (Snell *et al.*, 2017) first compute a class's prototype to be the mean of its support set in the embedding space. Then the transferability of feature embedding is evaluated by finding the nearest class prototype for embedded query points. An extension of prototypical networks is proposed in Ren *et al.* (2018) to deal with semi-supervised few-shot learning. Relation Network (Sung *et al.*, 2018) learns to learn a deep distance metric to compare a small number of images within episodes. Our proposed method is similar to these approaches in the sense that we all focus on learning deep embeddings with good generalization ability. However, our algorithm assumes a transductive setting, in which we utilize the union of support set and query set to exploit the manifold structure of novel class space by using episodic-wise parameters.

**Transduction** The setting of transductive inference was first introduced by Vapnik (Vapnik, 1999). Transductive Support Vector Machines (TSVMs) (Joachims, 1999) is a margin-based classification method that minimizes errors of a particular test set. It shows substantial improvements over inductive methods, especially for small training sets. Another category of transduction methods involves graph-based methods (Zhou *et al.*, 2004; Wang and Zhang, 2006; Rohrbach *et al.*, 2013; Fu *et al.*, 2015). Label propagation is used in Zhou *et al.* (2004) to transfer labels from labeled to unlabeled data instances guided by the weighted graph. Label propagation is sensitive to variance parameter $\sigma$, so Linear Neighborhood Propagation (LNP) (Wang and Zhang, 2006) constructs approximated Laplacian matrix to avoid this issue. In Zhu and Ghahramani (2002), minimum spanning tree heuristic and entropy minimization are used to learn the parameter $\sigma$. In all these prior work, the graph construction is done on a pre-defined feature space using manually selected hyperparamters since it is not possible to learn them at test time. Our approach, on the other hand, is able to learn the graph construction network since it is a meta-learning framework with episodic training, where at each episode we simulate the test set with a subset of the training set.

In few-shot learning, Nichol *et al.* (2018) experiments with a transductive setting and shows improvements. However, they only share information between test examples via batch normalization (Ioffe and Szegedy, 2015) rather than explicitly model the transductive setting as in our algorithm.

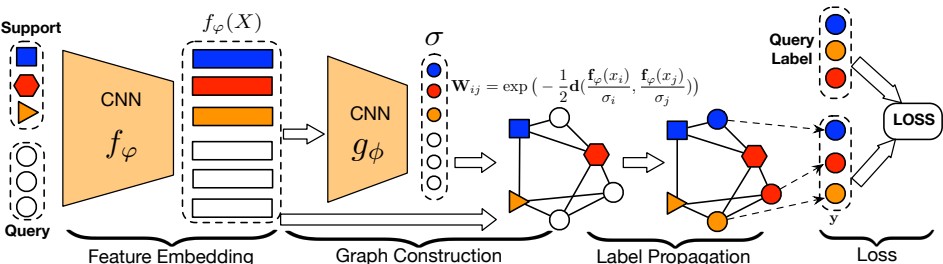

Figure 2: The overall framework of our algorithm in which the manifold structure of the entire query set helps to learn better decision boundary. The proposed algorithm is composed of four components: feature embedding, graph construction, label propagation, and loss generation.

## 3 MAIN APPROACH

In this section, we introduce the proposed algorithm that utilizes the manifold structure of the given few-shot classification task to improve the performance.

### 3.1 PROBLEM DEFINITION

We follow the episodic paradigm (Vinyals *et al.*, 2016) that effectively trains a meta-learner for few-shot classification tasks, which is commonly employed in various literature (Snell *et al.*, 2017; Finn *et al.*, 2017; Nichol *et al.*, 2018; Sung *et al.*, 2018; Mishra *et al.*, 2018). Given a relatively large labeled dataset with a set of classes $\mathcal{C}_{train}$, the objective of this setting is to train classifiers for an unseen set of novel classes $\mathcal{C}_{test}$, for which only a few labeled examples are available.

Specifically, in each episode, a small subset of $N$ classes are sampled from $\mathcal{C}_{train}$ to construct a *support set* and a *query set*. The *support set* contains $K$ examples from each of the $N$ classes (i.e., $N$-way $K$-shot setting) denoted as $\mathcal{S} = \{(\mathbf{x}_1, y_1), (\mathbf{x}_2, y_2), \ldots, (\mathbf{x}_{N \times K}, y_{N \times K})\}$, while the *query set* $\mathcal{Q} = \{(\mathbf{x}_1^*, y_1^*), (\mathbf{x}_2^*, y_2^*), \ldots, (\mathbf{x}_T^*, y_T^*)\}$ includes different examples from the same $N$ classes. Here, the support set $\mathcal{S}$ in each episode serves as the labeled training set on which the model is trained to minimize the loss of its predictions for the query set $\mathcal{Q}$. This procedure mimics training classifiers for $\mathcal{C}_{test}$ and goes episode by episode until convergence.

Meta-learning implemented by the episodic training reasonably performs well to few-shot classification tasks. Yet, due to the lack of labeled instances ($K$ is usually very small) in the support set, we observe that a reliable classifier is still difficult to be obtained. This motivates us to consider a transductive setting that utilizes the whole query set for the prediction rather than predicting each example independently. Taking the entire query set into account, we can alleviate the low-data problem and provide more reliable generalization property.

### 3.2 TRANSDUCTIVE PROPAGATION NETWORK (TPN)

We introduce *Transductive Propagation Network* (TPN) illustrated in Figure 2, which consists of four components: feature embedding with a convolutional neural network; graph construction that produces example-wise parameters to exploit the manifold structure; label propagation that spreads labels from the support set $\mathcal{S}$ to the query set $\mathcal{Q}$; a loss generation step that computes a cross-entropy loss between propagated labels and the ground-truths on $\mathcal{Q}$ to jointly train all parameters in the framework.

### 3.2.1 FEATURE EMBEDDING

We employ a convolutional neural network $f_\varphi$ to extract features of an input $\mathbf{x}_i$, where $f_\varphi(\mathbf{x}_i; \varphi)$ refers to the feature map and $\varphi$ indicates a parameter of the network. Despite the generality, we adopt the same architecture used in several recent works (Snell *et al.*, 2017; Sung *et al.*, 2018; Vinyals *et al.*, 2016). By doing so, we can provide more fair comparisons in the experiments, highlighting the effects of transductive approach. The network is made up of four convolutional blocks where each block begins with a 2D convolutional layer with a $3 \times 3$ kernel and filter size of $64$. Each

convolutional layer is followed by a batch-normalization layer (Ioffe and Szegedy, 2015), a ReLU nonlinearity and a $2 \times 2$ max-pooling layer. We use the same embedding function $f_\varphi$ for both the support set $\mathcal{S}$ and the query set $\mathcal{Q}$.

### 3.2.2 GRAPH CONSTRUCTION

Manifold learning (Chung and Graham, 1997; Zhou *et al.*, 2004; Yang *et al.*, 2016) discovers the embedded low-dimensional subspace in the data, where it is critical to choose an appropriate neighborhood graph. A common choice is Gaussian similarity function:

$$W_{ij} = \exp\left(-\frac{d(\mathbf{x}_i, \mathbf{x}_j)}{2\sigma^2}\right) , \tag{1}$$

where $d(\cdot, \cdot)$ is a distance measure (e.g., Euclidean distance) and $\sigma$ is the length scale parameter. The neighborhood structure behaves differently with respect to various $\sigma$, which means that it needs to carefully select the optimal $\sigma$ for the best performance of label propagation (Wang and Zhang, 2006; Zhu and Ghahramani, 2002). In addition, we observe that there is no principled way to tune the scale parameter in meta-learning framework, though there exist some heuristics for dimensionalty reduction methods (Zelnik-Manor and Perona, 2004; Sugiyama, 2007).

**Example-wise length-scale parameter**    To obtain a proper neighborhood graph in meta-learning, we propose a graph construction module built on the union set of support set and query set: $\mathcal{S} \cup \mathcal{Q}$. This module is composed of a convolutional neural network $g_\phi$ which takes the feature map $f_\varphi(\mathbf{x}_i)$ for $\mathbf{x}_i \in \mathcal{S} \cup \mathcal{Q}$ to produce an *example-wise* length-scale parameter $\sigma_i = g_\phi(f_\varphi(\mathbf{x}_i))$. Note that the scale parameter is determined example-wisely and learned in an episodic training procedure, which adapts well to different tasks and makes it suitable for few-shot learning. With the example-wise $\sigma_i$, our similarity function is then defined as follows:

$$W_{ij} = \exp\left(-\frac{1}{2}d\left(\frac{f_\varphi(\mathbf{x_i})}{\sigma_i}, \frac{f_\varphi(\mathbf{x_j})}{\sigma_j}\right)\right) \tag{2}$$

where $W \in R^{(N \times K + T) \times (N \times K + T)}$ for all instances in $\mathcal{S} \cup \mathcal{Q}$. We only keep the $k$-max values in each row of $W$ to construct a $k$-nearest neighbour graph. Then we apply the normalized graph Laplacians (Chung and Graham, 1997) on $W$, that is, $S = D^{-1/2} W D^{-1/2}$, where $D$ is a diagonal matrix with its $(i, i)$-value to be the sum of the $i$-th row of $W$.

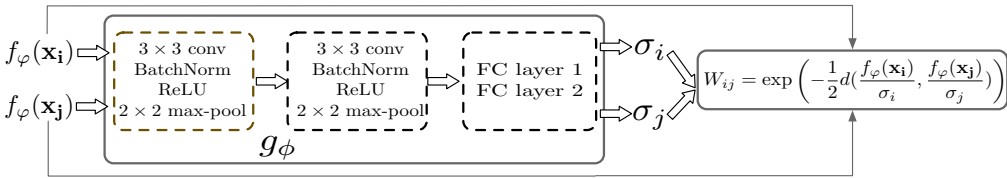

Figure 3: Detailed architecture of the graph construction module, in which the length-scale parameter is example-wisely determined.

**Graph construction structure**    The structure of the proposed graph construction module is shown in Figure 3. It is composed of two convolutional blocks and two fully-connected layers, where each block contains a 3-by-3 convolution, batch normalization, ReLU activation, followed by 2-by-2 max pooling. The number of filters in each convolutional block is 64 and 1, respectively. To provide an example-wise scaling parameter, the activation map from the second convolutional block is transformed into a scalar by two fully-connected layers in which the number of neurons is 8 and 1, respectively.

**Graph construction in each episode**    We follow the episodic paradigm for few-shot meta-learner training. This means that the graph is individually constructed for each task in each episode, as shown in Figure 1. Typically, in 5-way 5-shot training, $N = 5, K = 5, T = 75$, the dimension of $W$ is only $100 \times 100$, which is quite efficient.

### 3.2.3 LABEL PROPAGATION

We now describe how to get predictions for the query set $\mathcal{Q}$ using label propagation, before the last cross-entropy loss step. Let $\mathcal{F}$ denote the set of $(N \times K + T) \times N$ matrix with nonnegative entries. We define a label matrix $Y \in \mathcal{F}$ with $Y_{ij} = 1$ if $\mathbf{x}_i$ is from the support set and labeled as $y_i = j$, otherwise $Y_{ij} = 0$. Starting from $Y$, label propagation iteratively determines the unknown labels of instances in the union set $\mathcal{S} \cup \mathcal{Q}$ according to the graph structure using the following formulation:

$$F_{t+1} = \alpha S F_t + (1 - \alpha) Y, \tag{3}$$

where $F_t \in \mathcal{F}$ denotes the predicted labels at the timestamp $t$, $S$ denotes the normalized weight, and $\alpha \in (0, 1)$ controls the amount of propagated information. It is well known that the sequence $\{F_t\}$ has a closed-form solution as follows:

$$F^* = (I - \alpha S)^{-1} Y, \tag{4}$$

where $I$ is the identity matrix (Zhou *et al.*, 2004). We directly utilize this result for the label propagation, making a whole episodic meta-learning procedure more efficient in practice.

**Time complexity**    Matrix inversion originally takes $O(n^3)$ time complexity, which is inefficient for large $n$. However, in our setting, $n = N \times K + T$ (80 for 1-shot and 100 for 5-shot) is very small. Moreover, there is plenty of prior work on the scalability and efficiency of label propagation, such as Liang and Li (2018); Fujiwara and Irie (2014), which can extend our work to large-scale data. More discussions are presented in A.4

### 3.2.4 CLASSIFICATION LOSS GENERATION

The objective of this step is to compute the classification loss between the predictions of the union of support and query set via label propagation and the ground-truths. We compute the cross-entropy loss between predicted scores $F^*$ and ground-truth labels from $\mathcal{S} \cup \mathcal{Q}$ to learn all parameters in an end-to-end fashion, where $F^*$ is converted to probabilistic score using softmax:

$$P(\tilde{y}_i = j | \mathbf{x}_i) = \frac{\exp(F^*_{ij})}{\sum_{j=1}^N \exp(F^*_{ij})}. \tag{5}$$

Here, $\tilde{y}_i$ denotes the final predicted label for $i$th instance in the union of support and query set and $F^*_{ij}$ denotes the $j$th component of predicted label from label propagation. Then the loss function is computed as:

$$J(\varphi, \phi) = \sum_{i=1}^{N \times K+T} \sum_{j=1}^{N} -\mathbb{I}(y_i == j) \log(P(\tilde{y}_i = j | \mathbf{x}_i)), \tag{6}$$

where $y_i$ means the ground-truth label of $\mathbf{x}_i$ and $\mathbb{I}(b)$ is an indicator function, $\mathbb{I}(b) = 1$ if $b$ is true and 0 otherwise.

Note that in Equation (6), the loss is dependent on two set of parameters $\varphi$, $\phi$ (even though the dependency is implicit through $F^*_{ij}$). All these parameters are jointly updated by the episodic training in an end-to-end manner.

## 4 EXPERIMENTS

We evaluate and compare our TPN with state-of-the-art approaches on two datasets, i.e., *mini*ImageNet (Ravi and Larochelle, 2017) and *tiered*ImageNet (Ren *et al.*, 2018). The former is the most popular few-shot learning benchmark and the latter is a much larger dataset released recently for few-shot learning.

### 4.1 DATASETS

***mini*ImageNet**. The *mini*ImageNet dataset is a collection of Imagenet (Krizhevsky *et al.*, 2012) for few-shot image recognition. It is composed of 100 classes randomly selected from Imagenet with each class containing 600 examples. In order to directly compare with state-of-the-art algorithms for

few-shot learning, we rely on the class splits used by Ravi and Larochelle (2017), which includes 64 classes for training, 16 for validation, and 20 for test. All images are resized to $84 \times 84$ pixels.

***tiered*ImageNet**. Similar to *mini*ImageNet , *tiered*ImageNet (Ren *et al.*, 2018) is also a subset of Imagenet (Krizhevsky *et al.*, 2012), but it has a larger number of classes from ILSVRC-12 (608 classes rather than 100 for *mini*ImageNet). Different from *mini*ImageNet, it has a hierarchical structure of broader categories corresponding to high-level nodes in Imagenet. The top hierarchy has 34 categories, which are divided into 20 training (351 classes), 6 validation (97 classes) and 8 test (160 classes) categories. The average number of examples in each class is 1281. This high-level split strategy ensures that the training classes are distinct from the test classes semantically. This is a more challenging and realistic few-shot setting since there is no assumption that training classes should be similar to test classes. Similarly, all images are resized to $84 \times 84$ pixels.

## 4.2 EXPERIMENTAL SETUP

For fair comparison with other methods, we adopt a widely-used CNN (Finn *et al.*, 2017; Snell *et al.*, 2017) as the feature embedding function $f_\varphi$ (Section 3.2.1). The hyper-parameter $k$ of $k$-nearest neighbour graph (Section 3.2.2) is set to 20 and $\alpha$ of label propagation is set to 0.99, as suggested in Zhou *et al.* (2004).

Following Snell *et al.* (2017), we adopt the episodic training procedure, i.e, we sample a set of $N$-way $K$-shot training tasks to mimic the $N$-way $K$-shot test problems. Moreover, Snell *et al.* (2017) proposed a "Higher Way " training strategy which used more training classes in each episode than test case. However, we find that it is beneficial to train with more examples than test phase (Appendix A.1). This is denoted as "Higher Shot" in our experiments. For 1-shot and 5-shot test problem, we adopt 5-shot and 10-shot training respectively. In all settings, the query number is set to 15 and the performance are averaged over 600 randomly generated episodes from the test set.

All our models were trained with Adam (Kingma and Ba, 2015) and an initial learning rate of $10^{-3}$. For *mini*ImageNet, we cut the learning rate in half every $10,000$ episodes and for *tiered*ImageNet, we cut the learning rate every $25,000$ episodes. The reason for larger decay step is that *tiered*ImageNet has more classes and more examples in each class which needs larger training iterations. We ran the training process until the validation loss reached a plateau.

## 4.3 FEW-SHOT LEARNING RESULTS

We compare our method with several state-of-the-art approaches in various settings. Even though the transductive method has never been used explicitly, batch normalization layer was used transductively to share information between test examples. For example, in Finn *et al.* (2017); Nichol *et al.* (2018), they use the query batch statistics rather than global BN parameters for the prediction, which leads to performance gain in the query set. Besides, we propose two simple transductive methods as baselines that explicitly utilize the query set. First, we propose the MAML+Transduction with slight modification of loss function to: $\mathcal{J}(\theta) = \sum_{i=1}^{T} \mathbf{y}_i \log \mathbb{P}(\widehat{\mathbf{y}}_i | \mathbf{x}_i) + \sum_{i,j=1}^{N \times K+T} W_{ij} \| \widehat{\mathbf{y}}_i - \widehat{\mathbf{y}}_j \|_2^2$ for transductive inference. The additional term serves as transductive regularization. Second, the naive heuristic-based label propagation methods (Zhou *et al.*, 2004) is proposed to explicitly model the transductive inference.

Experimental results are shown in Table 1 and Table2. Transductive batch normalization methods tend to perform better than pure inductive methods except for the "Higher Way" PROTO NET. Label propagation without learning to propagate outperforms other baseline methods in most cases, which verifies the necessity of transduction. The proposed TPN achieves the state-of-the-art results and surpasses all the others with a large margin even when the model is trained with regular shots. When "Higher Shot" is applied, the performance of TPN continues to improve especially for 1-shot case. This confirms that our model effectively finds the episodic-wise manifold structure of test examples through learning to construct the graph for label propagation.

Another observation is that the advantages of 5-shot classification is less significant than that of 1-shot case. For example, in 5-way *mini*ImageNet , the absolute improvement of TPN over published state-of-the-art is 4.13% for 1-shot and 1.66% for 5-shot. To further investigate this, we experimented 5-way $k$-shot ($k = 1, 2, \cdots, 10$) experiments. The results are shown in Figure 4. Our TPN performs consistently better than other methods with varying shots. Moreover, it can be seen that

Table 1: Few-shot classification accuracies on *mini*ImageNet. All results are averaged over 600 test episodes. Top results are highlighted.

| Model | Transduction | 5-way Acc | | 10-way Acc | |
|---|---|---|---|---|---|
| | | 1-shot | 5-shot | 1-shot | 5-shot |
| **MAML (Finn *et al.*, 2017)** | BN | 48.70 | 63.11 | 31.27 | 46.92 |
| **MAML+Transduction** | Yes | 50.83 | 66.19 | 31.83 | 48.23 |
| **Reptile (Nichol *et al.*, 2018)** | No | 47.07 | 62.74 | 31.10 | 44.66 |
| **Reptile + BN (Nichol *et al.*, 2018)** | BN | 49.97 | 65.99 | 32.00 | 47.60 |
| **PROTO NET (Snell *et al.*, 2017)** | No | 46.14 | 65.77 | 32.88 | 49.29 |
| **PROTO NET (Higher Way) (Snell *et al.*, 2017)** | No | 49.42 | 68.20 | 34.61 | 50.09 |
| **RELATION NET (Sung *et al.*, 2018)** | BN | 51.38 | 67.07 | 34.86 | 47.94 |
| **Label Propagation** | Yes | 52.31 | 68.18 | 35.23 | 51.24 |
| **TPN** | Yes | **53.75** | **69.43** | **36.62** | **52.32** |
| **TPN (Higher Shot)** | Yes | **55.51** | **69.86** | **38.44** | **52.77** |

* "Higher Way" means using more classes in training episodes. "Higher Shot" means using more shots in training episodes. "BN" means information is shared among test examples using batch normalization.
† **Due to space limitation, we report the accuracy with 95% confidence intervals in Appendix.**

Table 2: Few-shot classification accuracies on *tiered*ImageNet. All results are averaged over 600 test episodes. Top results are highlighted.

| Model | Transduction | 5-way Acc | | 10-way Acc | |
|---|---|---|---|---|---|
| | | 1-shot | 5-shot | 1-shot | 5-shot |
| **MAML (Finn *et al.*, 2017)** | BN | 51.67 | 70.30 | 34.44 | 53.32 |
| **MAML + Transduction** | Yes | 53.23 | 70.83 | 34.78 | 54.67 |
| **Reptile (Nichol *et al.*, 2018)** | No | 48.97 | 66.47 | 33.67 | 48.04 |
| **Reptile + BN (Nichol *et al.*, 2018)** | BN | 52.36 | 71.03 | 35.32 | 51.98 |
| **PROTO NET (Snell *et al.*, 2017)** | No | 48.58 | 69.57 | 37.35 | 57.83 |
| **PROTO NET (Higher Way) (Snell *et al.*, 2017)** | No | 53.31 | 72.69 | 38.62 | 58.32 |
| **RELATION NET (Sung *et al.*, 2018)** | BN | 54.48 | 71.31 | 36.32 | 58.05 |
| **Label Propagation** | Yes | 55.23 | 70.43 | 39.39 | 57.89 |
| **TPN** | Yes | **57.53** | **72.85** | **40.93** | **59.17** |
| **TPN (Higher Shot)** | Yes | **59.91** | **73.30** | **44.80** | **59.44** |

* "Higher Way" means using more classes in training episodes. "Higher Shot" means using more shots in training episodes. "BN" means information is shared among test examples using batch normalization.
† **Due to space limitation, we report the accuracy with 95% confidence intervals in Appendix.**

TPN outperforms other methods with a large margin in lower shots. With the shot increase, the advantage of transduction narrows since more labelled data are used. This finding agrees with the results in TSVM (Joachims, 1999): when more training data are available, the bonus of transductive inference will be decreased.

## 4.4 Comparison with semi-supervised few-shot learning

Table 3: Semi-supervised comparison on *mini*ImageNet.

| Model | 1-shot | 5-shot | 1-shot w/D | 5-shot w/D |
|---|---|---|---|---|
| **Soft $k$-Means (Ren *et al.*, 2018)** | 50.09 | 64.59 | 48.70 | 63.55 |
| **Soft $k$-Means+Cluster (Ren *et al.*, 2018)** | 49.03 | 63.08 | 48.86 | 61.27 |
| **Masked Soft $k$-Means (Ren *et al.*, 2018)** | 50.41 | 64.39 | 49.04 | 62.96 |
| **TPN-semi** | **52.78** | **66.42** | **50.43** | **64.95** |

* "w/D" means with distraction. In this setting, many of the unlabelled data are from the so-called distraction classes , which is different from the classes of labelled data.
† **Due to space limitation, we report the accuracy with 95% confidence intervals in Appendix.**

The main difference of traditional semi-supervised learning and transduction is the source of unlabeled data. Transductive methods directly use test set as unlabeled data while semi-supervised learning usually has an extra unlabeled set. In order to compare with semi-supervised methods,

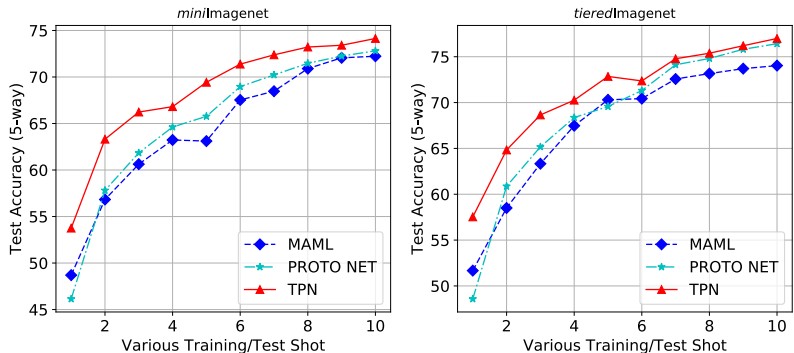

Figure 4: 5-way performance with various training/test shots.

Table 4: Semi-supervised comparison on *tiered*ImageNet.

| Model | 1-shot | 5-shot | 1-shot w/D | 5-shot w/D |
|---|---|---|---|---|
| **Soft $k$-Means (Ren *et al.*, 2018)** | 51.52 | 70.25 | 49.88 | 68.32 |
| **Soft $k$-Means+Cluster (Ren *et al.*, 2018)** | 51.85 | 69.42 | 51.36 | 67.56 |
| **Masked Soft $k$-Means (Ren *et al.*, 2018)** | 52.39 | 69.88 | 51.38 | 69.08 |
| **TPN-semi** | **55.74** | **71.01** | **53.45** | **69.93** |

\* "w/D" means with distraction. In this setting, many of the unlabelled data are from the so-called distraction classes , which is different from the classes of labelled data.
† **Due to space limitation, we report the accuracy with 95% confidence intervals in Appendix.**

we propose a semi-supervised version of TPN, named TPN-semi, which classifies one test example each time by propagating labels from the labeled set and extra unlabeled set.

We use *mini*ImageNet and *tiered*ImageNet with the labeled/unlabeled data split proposed by Ren *et al.* (2018). Specifically, they split the images of each class into disjoint labeled and unlabeled sets. For *mini*ImageNet, the ratio of labeled/unlabeled data is 40% and 60% in each class. Likewise, the ratio is 10% and 90% for *tiered*ImageNet. All semi-supervised methods (including TPN-semi) sample support/query data from the labeled set (e.g, 40% from *mini*ImageNet) and sample unlabeled data from the unlabeled sets (e.g, 60% from *mini*ImageNet). In addition, there is a more challenging situation where many unlabelled examples from other distractor classes (different from labelled classes).

Following Ren *et al.* (2018), we report the average accuracy over 10 random labeled/unlabeled splits and the uncertainty computed in standard error. Results are shown in Table 3 and Table 4. It can be seen that TPN-semi outperforms all other algorithms with a large margin, especially for 1-shot case. Although TPN is originally designed to perform transductive inference, we show that it can be successfully adapted to semi-supervised learning tasks with little modification. In certain cases where we can not get all test data, the TPN-semi can be used as an effective alternative algorithm.

# 5 CONCLUSION

In this work, we proposed the transductive setting for few-shot learning. Our proposed approach, namely *Transductive Propagation Network* (TPN), utilizes the entire test set for transductive inference. Specifically, our approach is composed of four steps: feature embedding, graph construction, label propagation, and loss computation. Graph construction is a key step that produces example-wise parameters to exploit the manifold structure in each episode. In our method, all parameters are learned end-to-end using cross-entropy loss with respect to the ground truth labels and the prediction scores in the query set. We obtained the state-of-the-art results on *mini*ImageNet and *tiered*ImageNet. Also, the semi-supervised adaptation of our algorithm achieved higher results than other semi-supervised methods. In future work, we are going to explore the episodic-wise distance metric rather than only using example-wise parameters for the Euclidean distance.

ACKNOWLEDGMENTS

Saehoon Kim, Minseop Park, and Eunho Yang were supported by Samsung Research Funding & Incubation Center of Samsung Electronics under Project Number SRFC-IT1702-15. Yanbin Liu and Yi Yang are in part supported by AWS Cloud Credits for Research.

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

# A   ABLATION STUDY

In this section, we performed several ablation studies with respect to training shots and query number.

## A.1   TRAINING SHOTS

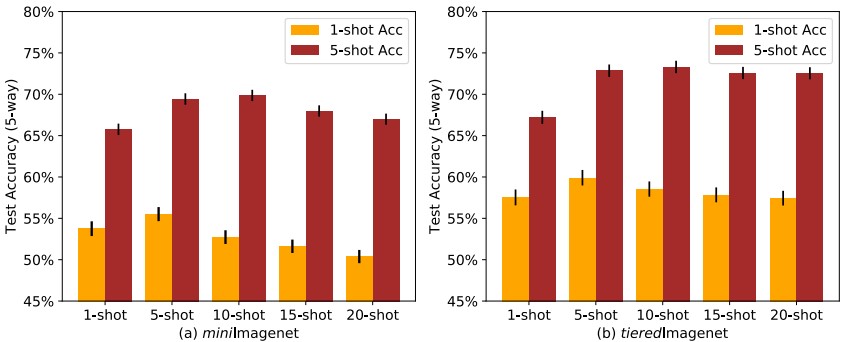

Figure 5: Model performance with different training shots. The $x$-axis indicates the number of shots in training, and the $y$-axis indicates 5-way test accuracy for 1-shot and 5-shot. Error bars indicate 95% confidence intervals as computed over 600 test episodes.

## A.2   QUERY NUMBER

Table 5: Accuracy with various query numbers

|  | *mini*ImageNet 1-shot | | | | | |
|---|---|---|---|---|---|---|
|  | 5 | 10 | 15 | 20 | 25 | 30 |
| Train=15 | 52.29 | 52.95 | 53.75 | 53.92 | 54.57 | 54.47 |
| Test=15 | 53.53 | 53.72 | 53.75 | 52.79 | 52.84 | 52.47 |
| Train=Test | 51.94 | 53.47 | 53.75 | 54.00 | 53.59 | 53.32 |
|  | *mini*ImageNet 5-shot | | | | | |
|  | 5 | 10 | 15 | 20 | 25 | 30 |
| Train=15 | 66.97 | 69.30 | 69.43 | 69.92 | 70.54 | 70.36 |
| Test=15 | 68.50 | 68.85 | 69.43 | 69.26 | 69.12 | 68.89 |
| Train=Test | 67.55 | 69.22 | 69.43 | 69.85 | 70.11 | 69.94 |

At first, we designed three experiments to study the influence of the query number in both training and test phase: (1) fix training query to 15; (2) fix test query to 15; (3) training query equals test query. The results are shown in Table 5. Some conclusions can be drawn from this experiment: (1) When training query is fixed, increasing the test query will lead to the performance gain. Moreover, even a small test query (e.g., 5) can yield good performance; (2) When test query is fixed, the performance is relatively stable with various training query numbers; (3) If the query number of training matches test, the performance can also be improved with increasing number.

## A.3   RESULTS ON RESNET

In this paper, we use a 4-layer neural network structure as described in Section 3.2.1 to make a fair comparison. Currently, there are two common network architectures in few-shot learning: 4-layer ConvNets (e.g., Finn *et al.* (2017); Snell *et al.* (2017); Sung *et al.* (2018)) and 12-layer ResNet (e.g., Mishra *et al.* (2018); Munkhdalai *et al.* (2018); Matthias *et al.* (2017); Oreshkin *et al.* (2018)). Our method belongs to the first one, which contains much fewer layers than the ResNet setting. Thus, it is more reasonable to compare algorithms such as TADAM (Oreshkin *et al.*, 2018) with ResNet version of our method. To make this comparison, we implemented our algorithm with ResNet architecture on miniImagenet dataset and show the results in Table 6.

It can be seen that we beat TADAM for 1-shot setting. For 5-shot, we outperform all other recent high-performance methods except for TADAM.

Table 6: ResNet results on *mini*ImageNet

| Method | 1-shot | 5-shot |
|---|---|---|
| SNAIL (Mishra *et al.*, 2018) | 55.71 | 68.88 |
| adaResNet (Munkhdalai *et al.*, 2018) | 56.88 | 71.94 |
| Discriminative k-shot (Matthias *et al.*, 2017) | 56.30 | 73.90 |
| TADAM (Oreshkin *et al.*, 2018) | 58.50 | **76.70** |
| TPN | **59.46** | 75.65 |

## A.4 CLOSED-FORM SOLUTION VS ITERATIVE UPDATES

There is a potential concern that the closed-form solution of label propagation can not scale to large-scale matrix. We relieve this concern from two aspects. On one hand, the few-shot learning problem assumes that training examples in each class is quite small (only 1 or 5). In this situation, Eq 3 and the closed-form version can be efficiently solved, since the dimension of $S$ is only $80 \times 80$ (5-way, 1-shot, 15-query) or $100 \times 100$ (5-way, 5-shot, 15-query). On the other hand, there are plenty of prior work on the scalability and efficiency of label propagation, such as Liang and Li (2018); Fujiwara and Irie (2014), which can extend our work to large-scale data.

Furthermore, on miniImagenet, we performed iterative optimization and got 53.05/68.75 for 1-shot/5-shot experiments with only 10 steps. This is slightly worse than closed-form version (53.75/69.43). We attribute this slightly worse accuracy to the inaccurate computation and unstable gradients caused by multiple step iterations.

## A.5 ACCURACY WITH 95% CONFIDENCE INTERVALS

Table 7: Few-shot classification accuracies on *mini*ImageNet. All results are averaged over 600 test episodes and are reported with 95% confidence intervals. Top results are highlighted.

| Model | Transduction | 5-way Acc | | 10-way Acc | |
|---|---|---|---|---|---|
| | | 1-shot | 5-shot | 1-shot | 5-shot |
| **MAML** | BN | 48.70±1.84 | 63.11±0.92 | 31.27±1.15 | 46.92±1.25 |
| **MAML+Transduction** | Yes | 50.83±1.85 | 66.19±1.85 | 31.83±0.45 | 48.23±1.28 |
| **Reptile** | No | 47.07±0.26 | 62.74±0.37 | 31.10±0.28 | 44.66±0.30 |
| **Reptile + BN** | BN | 49.97±0.32 | 65.99±0.58 | 32.00±0.27 | 47.60±0.32 |
| **PROTO NET** | No | 46.14±0.77 | 65.77±0.70 | 32.88±0.47 | 49.29±0.42 |
| **PROTO NET (Higher Way)** | No | 49.42±0.78 | 68.20±0.66 | 34.61±0.46 | 50.09±0.44 |
| **RELATION NET** | BN | 51.38±0.82 | 67.07±0.69 | 34.86±0.48 | 47.94±0.42 |
| **Label Propagation** | Yes | 52.31±0.85 | 68.18±0.67 | 35.23±0.51 | 51.24±0.43 |
| **TPN** | Yes | **53.75±0.86** | **69.43±0.67** | **36.62±0.50** | **52.32±0.44** |
| **TPN (Higher Shot)** | Yes | **55.51±0.86** | **69.86±0.65** | **38.44±0.49** | **52.77±0.45** |

\* "Higher Way" means using more classes in training episodes. "Higher Shot" means using more shots in training episodes. "BN" means information is shared among test examples using batch normalization.

Table 8: Few-shot classification accuracies on *tiered*ImageNet. All results are averaged over 600 test episodes and are reported with 95% confidence intervals. Top results are highlighted.

| Model | Transduction | 5-way Acc | | 10-way Acc | |
|---|---|---|---|---|---|
| | | 1-shot | 5-shot | 1-shot | 5-shot |
| **MAML** | BN | 51.67±1.81 | 70.30±1.75 | 34.44±1.19 | 53.32±1.33 |
| **MAML + Transduction** | Yes | 53.23±1.85 | 70.83±1.78 | 34.78±1.18 | 54.67±1.26 |
| **Reptile** | No | 48.97±0.21 | 66.47±0.21 | 33.67±0.28 | 48.04±0.30 |
| **Reptile + BN** | BN | 52.36±0.23 | 71.03±0.22 | 35.32±0.28 | 51.98±0.32 |
| **PROTO NET** | No | 48.58±0.87 | 69.57±0.75 | 37.35±0.56 | 57.83±0.55 |
| **PROTO NET (Higher Way)** | No | 53.31±0.89 | 72.69±0.74 | 38.62±0.57 | 58.32±0.55 |
| **RELATION NET** | BN | 54.48±0.93 | 71.31±0.78 | 36.32±0.62 | 58.05±0.59 |
| **Label Propagation** | Yes | 55.23±0.96 | 70.43±0.76 | 39.39±0.60 | 57.89±0.55 |
| **TPN** | Yes | **57.53±0.96** | **72.85±0.74** | **40.93±0.61** | **59.17±0.52** |
| **TPN (Higher Shot)** | Yes | **59.91±0.94** | **73.30±0.75** | **44.80±0.62** | **59.44±0.51** |

\* "Higher Way" means using more classes in training episodes. "Higher Shot" means using more shots in training episodes. "BN" means information is shared among test examples using batch normalization.

Table 9: Semi-supervised comparison on *mini*ImageNet.

| Model | 1-shot | 5-shot | 1-shot w/D | 5-shot w/D |
|---|---|---|---|---|
| **Soft $k$-Means** | 50.09±0.45 | 64.59±0.28 | 48.70±0.32 | 63.55±0.28 |
| **Soft $k$-Means+Cluster** | 49.03±0.24 | 63.08±0.18 | 48.86±0.32 | 61.27±0.24 |
| **Masked Soft $k$-Means** | 50.41±0.31 | 64.39±0.24 | 49.04±0.31 | 62.96±0.14 |
| **TPN-semi** | **52.78±0.27** | **66.42±0.21** | **50.43±0.84** | **64.95±0.73** |

\* "w/D" means with distraction. In this setting, many of the unlabelled data are from the so-called distraction classes , which is different from the classes of labelled data.

Table 10: Semi-supervised comparison on *tiered*ImageNet.

| Model | 1-shot | 5-shot | 1-shot w/D | 5-shot w/D |
|---|---|---|---|---|
| **Soft $k$-Means** | 51.52±0.36 | 70.25±0.31 | 49.88±0.52 | 68.32±0.22 |
| **Soft $k$-Means+Cluster** | 51.85±0.25 | 69.42±0.17 | 51.36±0.31 | 67.56±0.10 |
| **Masked Soft $k$-Means** | 52.39±0.44 | 69.88±0.20 | 51.38±0.38 | 69.08±0.25 |
| **TPN-semi** | **55.74±0.29** | **71.01±0.23** | **53.45±0.93** | **69.93±0.80** |

\* "w/D" means with distraction. In this setting, many of the unlabelled data are from the so-called distraction classes , which is different from the classes of labelled data.

