# OpenReview forum: "LEARNING TO PROPAGATE LABELS: TRANSDUCTIVE PROPAGATION NETWORK FOR FEW-SHOT LEARNING"
_ICLR.cc/2019/Conference_

### Official Review · AnonReviewer1 · 2018-10-27
**Transductive few-shot by meta-learning to propagate labels for . Solid work.**

**Rating:** 7
**Confidence:** 4

**Review:**

The paper studies few-host learning in a transductive setting: using meta learning to learn to propagate labels from training samples to test samples.

There is nothing strikingly novel in this work, using unlabeled test samples in a transductive way seem to help slightly. However, the paper does cover a setup that I am not aware that was studied before. The paper is written clearly, and the experiments seem solid.

Comments:
-- What can be said about how computationally demanding the procedure is? running label propagation within meta learning might be too costly.
-- It is not clear how the  per-example scalar sigma-i is learned. (for Eq 2)
-- solving Eq 3 by matrix inversion does not scale. Would be best to also show results using iterative optimization

---

> ### Author Response · Authors · 2018-11-25
> **Response to AnonReviewer1**
>
> Please refer to our main response in an above comment that addresses the primary and common questions amongst all reviewers. Here we respond to your specific comments.
>
> "What can be said about how computationally demanding the procedure is? running label propagation within meta learning might be too costly. "
>
> >>> In few-shot learning, episodic paradigm proposed by Matching Networks [1] is widely adopted by current researchers (we follow the same setting to make a fair comparison). In each episode, a small subset of N-way K-shot Q-query examples is sampled from the training set. Typically, for 1-shot experiments, N=5, K=1, Q=15 and for 5-shot experiments, N=5, K=5, Q=15. Thus, the number of training examples are Nx(K+Q) (80 for 1-shot and 100 for 5-shot). Constructing label propagation matrix W involves both support and query examples (80 or 100). So the dimension of W is either 80x80 or 100x100. Running label propagation on such small matrix is quite efficient.
>
> "It is not clear how the  per-example scalar sigma-i is learned. (for Eq 2)"
>
> >>> In Figure 4 of appendix A, we describe the detailed structure of the graph construction module. After we get the per-example feature representation f_{\varphi}(x_i) for x_i, we feed it into the graph construction module g_{\phi}. The output of this module is a one-dimensional scalar. f and g are learned in an end-to-end way in our approach.
>
> "solving Eq 3 by matrix inversion does not scale. Would be best to also show results using iterative optimization "
>
> >>> We want to answer this question from two aspects. On one hand, few-shot learning assumes that training examples in each class are quite small (only 1 or 5). In this situation, Eq (3) and the closed-form version can be efficiently solved, since the dimension of S is only 80x80 or 100x100. On the other hand, there is plenty of prior work on the scalability and efficiency of label propagation, such as [2], [3], [4], which can extend our work to large-scale data.
> On miniImagenet, we performed iterative optimization and got 53.05/68.75 for 1-shot/5-shot experiments with only 10 steps. This is slightly worse than closed-form version (53.75/69.43), because of the inaccurate computation and unstable gradients caused by multiple step iterations.
>
>
> [1] Vinyals, Oriol, et al. "Matching networks for one shot learning." NIPS. 2016.
> [2] Liang, De-Ming, and Yu-Feng Li. "Lightweight Label Propagation for Large-Scale Network Data." IJCAI. 2018.
> [3] Fujiwara, Yasuhiro, and Go Irie. "Efficient label propagation." ICML. 2014.
> [4] Weston, Jason. "Large-Scale Semi-Supervised Learning."

---

### Official Review · AnonReviewer2 · 2018-11-02
**Novel idea, but important details and deeper analysis are missing**

**Rating:** 6
**Confidence:** 3

**Review:**

Summary
This paper proposes a meta-learning framework that leverages unlabeled data by learning the graph-based label propogation in an end-to-end manner.  The proposed approaches are evaluated on two few-shot datasets and achieves the state-of-the-art results.

Pros.
-This paper is well-motivated. Studying label propagation in the meta-learning setting is interesting and novel. Intuitively, transductive label propagation should improve supervised learning when the number of labeled instances is low.
-The empirical results show improvement over the baselines, which are expected.

Cons.
-Some technical details  are missing. In Section 3.2.2, the authors only explain how they learn example-based \sigma, but details on how to make graph construction end-to-end trainable are missing. Constructing the full weight matrix requires the whole dataset as input and selecting k-nearest neighbor is a non-differentiable operation. Can you give more explanations?
-Does episode training help label propagation? How about the results of label propagation without the episode training?

---

> ### Author Response · Authors · 2018-11-25
> **Response to AnonReviewer2**
>
> Please refer to our main response in an above comment that addresses the primary and common questions amongst all reviewers. Here we respond to your specific comments.
>
> "Some technical details are missing. In Section 3.2.2, the authors only explain how they learn example-based \sigma, but details on how to make graph construction end-to-end trainable are missing. Constructing the full weight matrix requires the whole dataset as input and selecting k-nearest neighbor is a non-differentiable operation. Can you give more explanations?"
>
> >>> Thanks for pointing out the details. We want to clarify the few-shot setting. We follow the widely-used episodic paradigm proposed by Matching Networks [1]. In each episode (training batch), our algorithm solves a small classification problem which contains N classes each having K support and Q query examples (e.g., N=5, K=1, Q=15, totally 80 examples). The weight matrix is constructed on the support and query examples in each episode rather than the whole dataset. This is very fast and efficient.
> In deep neural networks, there is a common trick in computing the gradient of operations non-differentiable at some points, but differentiable elsewhere, such as Max-Pooling (top-1) and top-k. In forward computation pass, the index position of the max (or top-k) values are stored. While in the back propagation pass, the gradient is computed only with respect to these saved positions. This trick is implemented in modern deep learning frameworks such as tensorflow and pytorch. In our paper, we use the tensorflow function tf.nn.top_k() to compute k-nearest neighbor operation.
>
> "Does episode training help label propagation? How about the results of label propagation without the episode training? "
>
> >>> In our paper, the length scale parameter \sigma is trained in an example-wise and episodic-wise way, as described in section 3.2.2 and Figure 4 of Appendix A. In order to investigate the benefit of episodic training, we combine the heuristic-based label propagation methods [2] with meta-learning to serve as a transductive baseline. Please refer to Table 1 and Table 2 line "Label Propagation". It can be seen that TPN outperforms naive label propagation with a large margin, thus verifying the effectiveness of episode training.
>
>
> [1] Vinyals, Oriol et al. "Matching networks for one shot learning." NIPS. 2016.
> [2] Zhou, Denny et al. "Learning with local and global consistency." NIPS. 2004.

---

### Official Review · AnonReviewer3 · 2018-11-03
**interesting empirically**

**Rating:** 5
**Confidence:** 3

**Review:**

This paper proposes to address few-shot learning in a transductive way by learning a label propagation model in an end-to-end manner.  Semi-supervised few-shot learning is important considering the limitation of the very few labeled instances. This is an interesting work.

The merits of this paper lie in the following aspects: (1) It is the first to learn label propagation for transductive few-shot learning. (2) The proposed approach produced effective empirical results.

The drawbacks  of the work include the following: (1) There is not much technical contribution. It merely just puts the CNN representation learning and the label propagation together to perform end-to-end learning. Considering the optimization problem involved in the learning process, it is hard to judge whether the effect of such a procedure from the optimization perspective.  (2) Empirically, it seems TPN achieved very small improvements over the very baseline label propagation.  Moreover, the performance reported in this paper seems to be much inferior to the state-of-the-art results reported in the literature. For example,  on miniImageNet, TADAM(Oreshkin et al, 2018) reported 58.5 (1-shot) and 76.7(5-shot), which are way better than the results reported in this work. This is a major concern.

---

> ### Author Response · Authors · 2018-11-25
> **Response to AnonReviewer3**
>
> Please refer to our main response in an above comment that addresses the primary and common questions amongst all reviewers. Here we respond to your specific comments.
>
> "(1) There is not much technical contribution. It merely just puts the CNN representation learning and the label propagation together to perform end-to-end learning. Considering the optimization problem involved in the learning process, it is hard to judge whether the effect of such a procedure from the optimization perspective."
>
> >>> As mentioned in the main response, the proposed TPN is not a mere combination of CNN representation learning and label propagation. The original label propagation constructs a fixed graph (Eq (1)) to explore the correlation between examples. While in our work, we adaptively construct the graph structure for each episode (training task) with a learnable graph construction module (Figure 4, Appendix A). This leads to better generalization ability for test tasks.
> In Table 1 and Table 2, the proposed TPN achieved much higher accuracy than the mere combination model (referred to as "Label Propagation").
>
> "(2) Empirically, it seems TPN achieved very small improvements over the very baseline label propagation.  Moreover, the performance reported in this paper seems to be much inferior to the state-of-the-art results reported in the literature. For example,  on miniImageNet, TADAM(Oreshkin et al, 2018) reported 58.5 (1-shot) and 76.7(5-shot), which are way better than the results reported in this work. This is a major concern."
>
> >>> At first, we want to clarify the few-shot network architecture setting. Currently, there are two common network architectures: 4-layer ConvNets (e.g., [1][2][3]) and 12-layer ResNet (e.g., [4][5][6][7]). Our method belongs to the first one, which contains much fewer layers than the ResNet setting. Thus, it is more reasonable to compare TADAM with ResNet version of our method. To better relieve the reviewer's concern, we implemented our algorithm with ResNet architecture on miniImagenet dataset and show the results as follow:
>
> Method                                  1-shot    5-shot
> SNAIL [4]                                 55.71     68.88
> adaResNet [5]                        56.88     71.94
> Discriminative k-shot [6]     56.30     73.90
> TADAM [7]                              58.50     76.70
> --------------------------------------------------------
> Ours                                        59.46     75.65
> --------------------------------------------------------
>
> It can be seen that we beat TADAM for 1-shot setting. For 5-shot, we outperform all other recent high-performance methods except for TADAM.
>
> >>> We want to clarify that "Label Propagation" in Table 1 and Table 2 is a strong baseline. It combines label propagation method [8] with episodic meta-learning. The usage of transductive inference makes this baseline outperform most published state-of-the-art methods. Moreover, the performance of TPN over label propagation is not very small. For example, in miniImagenet, TPN outperforms label propagation with 1.44% and 1.25% for 1-shot and 5-shot respectively, but this advantage grows to 3.20% and 1.68% with "Higher Shot" training. The improvements are even larger for tieredImagenet with 4.68% and 2.87%. We believe in few-shot learning, this is a large improvement.
>
>
> [1] Finn, Chelsea, Pieter Abbeel, and Sergey Levine. "Model-agnostic meta-learning for fast adaptation of deep networks." ICML. 2017.
> [2] Snell, Jake, Kevin Swersky, and Richard Zemel. "Prototypical networks for few-shot learning." NIPS. 2017.
> [3] Yang, Flood Sung Yongxin et al. "Learning to compare: Relation network for few-shot learning." CVPR. 2018.
> [4] Mishra, Nikhil et al. "A simple neural attentive meta-learner." ICLR. 2018.
> [5] Munkhdalai, Tsendsuren et al. "Rapid adaptation with conditionally shifted neurons." ICML. 2018.
> [6] Bauer, Matthias et al. "Discriminative k-shot learning using probabilistic models." arXiv. 2017.
> [7] Oreshkin, B.N., Lacoste, A. and Rodriguez, P., 2018. "TADAM: Task dependent adaptive metric for improved few-shot learning." NIPS. 2018.
> [8] Zhou, Denny, et al. "Learning with local and global consistency." NIPS. 2004.

---

### Public Comment · (anonymous) · 2018-10-01
**Reasonable and efficacy idea**

This paper proposes a novel meta-learning framework, which aims to propagate labels from labeled instances to unlabeled test instances. This framework learns a graph construction module, exploiting the manifold structure in the data. The idea is reasonable and efficacy, and the experiments are comprehensive.

---

### Public Comment · (anonymous) · 2018-10-01
**Pointer to a Related Work**

 The paper looks very interesting as transductive approaches are powerful for metric learning and semi-supervised learning. And, learning to transductive learn is an interesting direction. I would like to point a related work which authors probably missed which performs metric learning/transfer learning using transduction: https://papers.nips.cc/paper/6360-learning-transferrable-representations-for-unsupervised-domain-adaptation

---

> ### Author Response · Authors · 2018-10-02
> **Thank you for pointing out related work**
>
> Thanks for pointing out the related work. We would like to include this reference to our manuscript in the next version.
>
> Our paper and the mentioned paper share the same idea of using metric learning and transduction. However, the target tasks are different. We focus on few-shot learning and meta-learning while the mentioned paper deals with unsupervised domain adaptation. This distinction leads to different algorithm designs: we learn to propagate labels while the mentioned paper proposes the transduction and adaptation steps.

---

### Public Comment · (anonymous) · 2018-10-02
**About the experiments**

It is interesting that this paper use a label propagation way to solve the low-data testing problem. However, the state-of-art few-shot(zero-shot) methods: Relation Net and Prototypical Net used both minImageNet, Omniglot for few-shot Testing and CUB-200 for zero-shot. So what's your results on Omniglot since you follow the idea of Prototypical Net. In addition, is it possible that your proposed TPN can deal with zero-shot problems since a general few-shot framework can Easily extend to cope with zero-shot problems?

---

> ### Author Response · Authors · 2018-10-08
> **Response to the experiments**
>
> Thanks for the comments.
> The state-of-the-art performance on Omniglot is quite high (>99% except for 20way-1shot setting), which means this problem is nearly solved. Also, there is a tendency that recent high-quality papers do not report results on Omniglot, such as TADAM [1] (NIPS2018), Delta-encoder [2] (NIPS2018), LEO [3] (ICLR19 submission). For the 20way-1shot setting, we compare our TPN results with Relation Net and Prototypical Net as follows:
> 					20way-1shot
> Prototypical Net           96.00
> Relation Net                  97.60
> TPN				        98.03
>
> Although zero-shot learning is not our focus, TPN can be easily adapted to zero-shot setting. The modification is similar to Prototypical network or Relation Network. First, a function g can be used to map class-level semantic feature into the same space of visual feature. Then, we can construct graph structure using both features and perform label propagation as in few-shot setting.
>
> [1] Oreshkin, Boris N., Alexandre Lacoste, and Pau Rodriguez. "TADAM: Task dependent adaptive metric for improved few-shot learning." NIPS2018
> [2] Schwartz, Eli, et al. "Delta-encoder: an effective sample synthesis method for few-shot object recognition." NIPS2018
> [3] Anonymous, "Meta-Learning with Latent Embedding Optimization." ICLR2019 submission.

---

### Public Comment · (anonymous) · 2018-10-13
**Experiments and Methods**

This paper tried to introduce transductive networks for few-shot learning.
I want to know about the experiments here especially about the transductive process that was done for query set. I hope that you can make it clear what was specifically performed in the batch of query set to help you gain the performance?
Do you also have any results if you increase the number of query set will affect your performance too? Because I believe this is the contribution that you can have as well from your work.

One more thing:
I have a question in your results for semisupervised few-shot learning.
I read the experiments in semisupervised few-shot learning protocol[1] that there are distractor classes in which I did not see this thing in your paper.
I intuitively think that this method might be appropriate for the unlabeled data without many outliers/distractors.
Do you also have the experiments about this before? It is  fine if you also show the drawback of this method, so the improvements can be proposed in the future to tackle that problem.




[1] Mengye Ren, Eleni Triantafillou, Sachin Ravi, Jake Snell, Kevin Swersky, Joshua B Tenenbaum, Hugo
Larochelle, and Richard S Zemel. Meta-learning for semi-supervised few-shot classification. International Conference on Learning Representations, 2018.

---

> ### Author Response · Authors · 2018-10-22
> **Response to experiments and methods**
>
> Thanks for the comments.
>
> For each episode, we first utilize both the support set and query set to construct the graph structure. Then, label propagation is performed according to the graph information to get all query set labels. The performance gain comes from the fact that we share information among all query examples and learn to propagate labels. In contrast, inductive methods predict query examples one by one, which does not enjoy this benefit.
>
> As to query set number experiments, please refer to Appendix B.2 for detailed information.
>
> For distractor classes, this is not the main focus of our paper. However, in order to explore the extent of our method, we performed experiments in the presence of distractor classes (same setting as [1]). The results are shown below:
> Model                                  mini-5way1shot     mini-5way5shot    tiered-5way1shot    tiered-5way5shot
> Soft k-Mean [1]                     48.70+/-0.32            63.55+/-0.28           49.88+/-0.52            68.32+/-0.22
> Soft k-Mean+Cluster [1]      48.86+/-0.32            61.27+/-0.24           51.36+/-0.31            67.56+/-0.10
> Masked Soft k-Means [1]    49.04+/-0.31            62.96+/-0.14           51.38+/-0.38            69.08+/-0.25
> TPN-semi (Ours)                   50.43+/-0.84            64.95+/-0.73           53.45+/-0.93            69.93+/-0.80
>
> It can be seen that our TPN-semi algorithm outperforms [1] in all cases, although our method is not specifically designed for the distractor-classes problem.
> We believe with care design, the performance of our method will continue to increase. This will be the future work.

---

### Public Comment · ~anon_ml_reviewer1 · 2018-11-15
**Reproducibility issues**

Since the paper has not provided a reproducible code, based on my implementation in PyTorch 1.0.0.dev20181105, unfortunately I could not reproduce their results on Mini-Imagenet dataset for 5-way, 1-shot and 5-shot scenarios. Using the exact mentioned hyper-parameters, the model didn't learn much in the end-to-end manner and the test accuracy for 5-way, 1-shot was around 25% trained in 50,000 episodes and learning-rate is halved every 10,000 episodes. Instead I pretrained the emebedding in the train set and decreased alpha (label propagation) to 0.9 then it started learning better.

The best accuracy I could get for the 5-way, 1-shot case (trained with 5-way, 1-shot so no higher-shot) is with alpha=0.6 and it is 47.93% (+/- 1.14% as the 95% confidence interval) which is much lower than the claimed 53.75%. More precisely, I trained with batch-size=15 as described in RelationNet, k in knn=20, Xavier initialization of Conv layers, BatchNorm unit weight initialization and zero bias, zero-mean normal initialization with std 0.01 for Linear layers and unit bias, and then tested with 15 query examples where results were averaged over 600 randomly generated episodes from the test set.

The paper did not mention any pre-processing step, so I only resized images to 84 by 84 and normalized the mini-imagenet data using imagenet mean and std.

---

> ### Author Response · Authors · 2018-11-16
> **Implementation details. Code will be released soon.**
>
> Thanks for the comment and interest about our paper.
> According to the blind review policy, we can not release the code at this moment. We will release our code and the trained model as soon as the review process ends. Meanwhile, we have sent an email to the program chairs to check if it is allowed to release the code anonymously.  We will share the code upon approval.
>
> We are sure about the reproducibility of the results shown in our paper. And in order to ensure the reproducibility, we ran the test procedure 10 times (each with 600 randomly generated episodes) and reported the average results to avoid accidentally high results. We are not sure if you have reproduced the result as outlined in [1]. If not, we sincerely hope you first try to reproduce the baseline method [1], so you may be closer to the right implementation. It took us quite a while to reproduce [1] even the code has been released.
> Nevertheless, we would like to provide more details below which could be useful for you to reproduce the results of our paper.
>
> (1) Our implementation is based on Tensorflow 1.3+, and we also tested on Pytorch 0.4.0. There is only a slight accuracy difference.
> (2) The reason why your results only achieved 25% could be caused by value issues such as divided by zero. Sincerely hope you could double check your code and please make sure you have added an epsilon whenever you call a divide operation.
> (3) Our model is learned end-to-end from scratch, and no pretrain is needed. We did not see your code, but we reckon you did not use the validation set to decide the early stopping iteration, which is commonly used in few-shot learning, such as Prototypical networks. Please use this practice if it is the case.
> (4) The detailed hyperparameters are: alpha=0.99, k=20, query=15, lr=0.001 and halved every 10,000 episodes for at most 100,000 episodes.
> (5) Network architecture details: feature extraction module is exactly the same as Prototypical networks [1], graph construction module is described in Figure 4 of Appendix A. Note that BatchNorm is applied only in Conv layers. In Figure 4, there is no Relu activation after FC layer2. More training details: we use Tensorflow default initialization, BatchNorm with default parameters: decay=0.999 and epsilon=0.001.
> (6) As to preprocessing, for miniImagenet, we follow Prototypical networks [1] while for tieredImagnet we follow Ren et al. [2].
>
> We have endeavored our best to 'guess' what mistakes you may have made, but there could be other issues that we are unable to enumerate.
> We highly suggest that a basic starting point is to reproduce the results of Prototypical networks.  Below we provide a few good implementation codes of some related papers.
> Prototypical networks:
> 	https://github.com/jakesnell/prototypical-networks
> 	https://github.com/cyvius96/prototypical-network-pytorch
> tieredImagenet:
> 	https://github.com/renmengye/few-shot-ssl-public
>
>
> [1] Snell, Jake, Kevin Swersky, and Richard Zemel. "Prototypical networks for few-shot learning." NIPS. 2017.
> [2] Ren, Mengye, et al. "Meta-learning for semi-supervised few-shot classification." ICLR. 2018.

---

> > ### Public Comment · ~anon_ml_reviewer1 · 2018-11-17
> > **Problem persisted**
> >
> > Thank you for the clarifications! I have already used Snell's code for the embedding and admit baseline implementation is tricky. I have not used Snell's pre-processing as the github repository contains the pre-processing for omniglot only, instead used the RelationNet data pre-processing and loading for mini-imagenet for few corrections such as the normalization mean, std from imagenet https://github.com/floodsung/LearningToCompare_FSL
> >
> > I have double checked the hyper-parameters and accessed the values in debug mode extensively and wherever division happens, I added an epsilon of 1e-6 or 1e-8 including the element-wise division of f_phi(x_i) / (sigma_i + epsilon) and in the computation of D^{-1/2} where I previously used torch.rsqrt() and replaced it with 1.0 / (w.sum(1) + epsilon).sqrt(), (where w is the graph knn matrix with applied masked k_max=20 for each row and zero everywhere else). However, the issue persisted with alpha=0.99 and model does not learn. As said previously, changing alpha to 0.9 or even lower 0.6 helped learning a lot but the final accuracy for 5-way, 1-shot case remained around the previous result of 47.9%.
> >
> > I hope ICLR authorities take anonymous code release into the considerations as this is a major barrier for assessing the reproducibility.

---

> > > ### Author Response · Authors · 2018-11-25
> > > **Anonymous Code Link**
> > >
> > > Thanks for the feedback.
> > >
> > > We got the approval from program chairs to release an anonymous code link, as follow:
> > > https://github.com/anonymisedsupplemental/TPN
> > >
> > > We would like to answer the related questions about our paper and code.

---

> > > > ### Public Comment · ~anon_ml_reviewer1 · 2018-11-28
> > > > **Issue found**
> > > >
> > > > Hi
> > > >
> > > > Thank you for the code!
> > > >
> > > > Please see my comment below: https://openreview.net/forum?id=SyVuRiC5K7&noteId=HkgdqB_y1V
> > > >
> > > > One suggestion; it'd be very helpful if you could add explicitly the pre-processing steps that was used to get the pickled data since I could not find it in "Meta-Learning for Semi-Supervised Few-Shot Classification" by Mengye Ren et. al.

---

> > > > > ### Author Response · Authors · 2018-11-28
> > > > > **Thanks for the reproduction. Pre-processing details.**
> > > > >
> > > > > Thanks for the clarification of reproduction.
> > > > >
> > > > > For the pre-processing step you mentioned, here we have two pieces of advices:
> > > > > 1. We have the dataset flag '--pkl' which controls the usage of pkl data or original image data (our own preprocessing). We tested with our own preprocessing, the performance is similar to pkl data.
> > > > > 2. For Mengye Ren's code, I think you can refer to https://github.com/renmengye/few-shot-ssl-public/blob/master/fewshot/data/mini_imagenet.py for more details.
> > > > >
> > > > > For other code issues, we are glad to offer help in github issue.

---

> > > > > > ### Public Comment · ~anon_ml_reviewer1 · 2018-12-01
> > > > > > **Issue found: Disparity of CELoss in paper and the code**
> > > > > >
> > > > > > Hi
> > > > > >
> > > > > > In section 3.2.4, it was written that cross-entropy loss is computed between F* and query labels, however, https://github.com/anonymisedsupplemental/TPN/blob/master/models.py#L145 the loss is computed between F* and the UNION of support labels and query labels. In fact, I changed the loss computation in your code to only use query labels and it resulted in poor accuracy similar to my earlier findings and decreasing alpha would match my previous results.
> > > > > >
> > > > > > This is the main issue I've had when my implementation did not work even after some compatible initializations between tensorflow and pytorch.
> > > > > >
> > > > > > Also for completeness, please add the relu after FC layer 1 in Figure 4 for graph construction.

---

> > > > > > > ### Author Response · Authors · 2018-12-03
> > > > > > > **Thanks for the feedback. New version will be revised.**
> > > > > > >
> > > > > > > Thanks for the feedback.
> > > > > > >
> > > > > > > We are going to revise the paper according to the useful suggestions regarding ce-loss and Figure4.

---

### Author Response · Authors · 2018-11-25
**Main response to reviewers**

We wish to thank the reviewers for their enlightening feedback! We would like to highlight the novelty and contribution of the proposed method.

(1) The proposed TPN is not a direct combination of the original label propagation and few-shot learning, but a novel transductive meta-learning method when facing unevenly distributed data.

The main contribution of the proposed TPN is to propose a novel transductive meta-learning method when facing an uneven data distribution. Most of previous (label) propagation algorithms usually assume samples are distributed evenly in the data space. Unfortunately, in low-shot learning, the data is limited and unevenly distributed, which makes most of existing label propagation algorithms inapplicable. To clearly model the data distribution in low-shot learning settings, previous transductive methods adopt a fixed scheme to explore the correlations between data, i.e., compute the weights with a fixed \sigma as shown in Eq (1). However, as pointed out in previous work [2][3] and in our experimental results, the performance of a transductive method is quite sensitive to the parameter \sigma, and a fixed \sigma will lead to suboptimal results. Our proposed TPN adaptively learns data correlation by calculating an optimal \sigma on a per data basis. As shown in Figure 4 (Appendix A), the correlation among data pairs is optimized and updated in each episode according to data distribution of the neighborhood. In this way, a different model is specifically learned to uncover the correlation of each data pair, thereby largely ameliorating the uneven data distribution problem.
Experimentally, TPN shows a big advantage over the direct combination (referred to as "Label Propagation" in Table 1 and Table 2).

(2) To the best of our knowledge, we are the first to model transductive inference explicitly in the few-shot meta-learning. This transductive setting paves a new way to solve the limited data problem in few-shot learning. As shown in this paper, if one has the test data in whole or a batch manner, transductive inference significantly improves the performance without additional human annotations.

(3) We advanced the state-of-the-art performance on the two most commonly-used benchmark datasets with large margins using the standard 4-layer ConvNets architecture.



[1] Zhou, Denny, et al. "Learning with local and global consistency." NIPS. 2004.
[2] Wang, Fei, and Changshui Zhang. "Label propagation through linear neighborhoods." TKDE. 2008.
[3] Xiaojin Z, Zoubin G. "Learning from labeled and unlabeled data with label propagation." Technical Report. 2002.

---

### Meta-Review · Area_Chair1 · 2018-12-13
**A new transductive few-shot learning algorithm with strong empirical results**

**Confidence:** 4
**Recommendation:** Accept (Poster)

**Metareview:**

As far as I know, this is the first paper to combine transductive learning with few-shot classification. The proposed algorithm, TPN, combines label propagation with episodic training, as well as learning an adaptive kernel bandwidth in order to determine the label propagation graph. The reviewers liked the idea, however there were concerns of novelty and clarity. I think the contributions of the paper and the strong empirical results are sufficient to merit acceptance, however the paper has not undergone a revision since September. It is therefore recommended that the authors improve the clarity based on the reviewer feedback. In particular, clarifying the details around learning \sigma_i and graph construction. It would also be useful to include the discussion of timing complexity in the final draft.